# Application of *meso*-CF_3_-Fluorophore BODIPY with Phenyl and Pyrazolyl Substituents for Lifetime Visualization of Lysosomes

**DOI:** 10.3390/molecules27155018

**Published:** 2022-08-07

**Authors:** Irina S. Trukhan, Denis N. Tomilin, Nataliya N. Dremina, Lyubov N. Sobenina, Michael G. Shurygin, Konstantin B. Petrushenko, Igor K. Petrushenko, Boris A. Trofimov, Irina A. Shurygina

**Affiliations:** 1Irkutsk Scientific Center of Surgery and Traumatology, 664003 Irkutsk, Russia; 2A.E. Favorsky Irkutsk Institute of Chemistry, Siberian Branch, Russian Academy of Science, 664033 Irkutsk, Russia; 3Irkutsk National Research Technical University, 664074 Irkutsk, Russia

**Keywords:** BODIPY, fluorescent sensor, Ehrlich ascites carcinoma, lysosome, BioStation CT

## Abstract

A bright far-red emitting unsymmetrical *meso*-CF_3_-BODIPY fluorescent dye with phenyl and pyrazolyl substituents was synthesized by condensation of trifluoropyrrolylethanol with pyrazolyl-pyrrole, with subsequent oxidation and complexation of the formed dipyrromethane. This BODIPY dye exhibits optical absorption at λ_ab_ ≈ 610–620 nm and emission at λ_em_ ≈ 640–650 nm. The BODIPY was studied on Ehrlich carcinoma cells as a lysosome-specific fluorescent dye that allows intravital staining of cell structures with subsequent real-time monitoring of changes occurring in the cells. It was also shown that the rate of uptake by cells, the rate of intracellular transport into lysosomes, and the rate of saturation of cells with the dye depend on its concentration in the culture medium. A concentration of 5 μM was chosen as the most suitable BODIPY concentration for fluorescent staining of living cell lysosomes, while a concentration of 100 μM was found to be toxic to Ehrlich carcinoma cells.

## 1. Introduction

Boradiazaindacene (BODIPY) dyes are now becoming more and more widespread in applied sciences among a large family of applicable fluorescent dyes, due to their unique properties. Their photophysical characteristics make it possible to use BODIPY-based materials in dye-sensitized solar cells [1,2,3], OLEDs and dye lasers [4], for photo-catalytic hydrogen generation [5], nonlinear optics [6,7], naked-eye metal detection [8,9], and sensors of biological targets [10,11,12].

BODIPY-based fluorescent probes play a key role in biotechnological and biochemical research, allowing visualization and tracking of biologically significant molecules, drugs, and macrostructures [13,14], which makes them a powerful tool for studying intercellular and intracellular biochemical processes; for example, for evaluating the medical efficacy and cytotoxic effects of a substance in the process of preclinical development of biologically active compounds [15,16,17]. The significance of BODIPY-based probes for diagnostic studies was demonstrated in a work in which an activity-based probe, bearing a BODIPY motif, was able to visualize active SARS-CoV-2 main protease within the nasopharyngeal epithelial cells of patients with active COVID-19 infection [18].

Among the known probes with absorption/emission peaks in the 300–800 nm range, those with near-infrared (NIR) absorption and emission are considered to be preferable due to their better tissue penetration, and lower interference and photo damage to living cells [19]. BODIPY-based probes meet this requirement and have proven their efficacy among commercially available substances for fluorescent imaging in the healthcare industry and allied fields [13], which are actively using this effective and versatile method for non-invasive in vivo and in vitro biochemical real-time research [20].

Lysosome imaging is of special interest, since they are involved in many physiological and pathological processes occurring in living cells, such as macromolecule degradation, plasma membrane repair, cellular homeostasis maintenance, apoptosis, and destruction product removal [21,22,23,24,25]. Participating in signal transduction, lysosomes contribute to proliferation, anabolic and catabolic process regulation, and also affect the activity of other cellular compartments and are involved in the cell immune response [26,27]. In this regard, hereditary or acquired lysosome dysfunctions are responsible, not only for a group of disorders of lysosome accumulation, but also play a significant role in the development and progression of certain diseases associated with metabolic disorders, including cancer [28,29,30,31].

Thus, studies of pathological processes related to lysosome function damage, including carcinogenesis, metastasis, and drug resistance of tumor cells, determined the importance of real-time lysosome imaging at the subcellular level for developing strategies aimed at suppressing tumor growth and enhancing antitumor therapy [30,32,33].

For lysosome staining, the most common tools are LysoTracker/LysoSensor (ThermoFisher Scientific, Waltham, MA, USA) and LYSO-ID (Enzo Biochem Inc, New York, NY, USA) widely used for biochemical investigations [34]. Unfortunately, these dyes are not without flaws: storage requires low temperature and light protection, and application is limited only to short-term visualization [35]. In addition, it was reported, that LysoTracker was inclined to reversible photoconversion to green after illumination by a light source equipped with a 560/40 excitation filter [36] and could act as P-glycoprotein transport substrate [37], which should be taken into account when using these probes.

In this regard, the search for fluorescent dyes capable of selectively staining lysosomes, suitable for use in living cell cultures, readily soluble in water media, and non-toxic and stable inside cells under experimental conditions and during storage remains a relevant task.

## 2. Results and Discussion

### 2.1. Synthesis of BODIPY **1**

Herein, the synthesis of BODIPY dye **1**, 4,4-difluoro-5-phenyl-3-(1,5-diphenyl-3-1*H*-pyrazolyl)-8-trifluoromethyl-4-boron-3a,4a-diaza-*s*-indacene as a potential intravital lysosome sensor is reported.

The choice of this compound was due to the fact that BODIPY fluorophores containing aromatic or heterocyclic substituents at the 3- and (or) 5-positions [38,39] or the strongly electron-withdrawing substituents, such as CF_3_, at the *meso*-position of BODIPY core, cause a deep bathochromic shift of absorption and emission maxima and therefore can be applied in biochemical studies. For example, *meso*-CF_3_-BODIPY dyes are employed as fluorescent sensors for in vivo imaging systems and in photodynamic therapy [40,41,42,43,44,45,46]. The BODIPY dyes with a *meso*-CF_3_-group also have at least two benefits as biochemical probes. First, the probe molecule is small. Second, the CF_3_-group is known to be useful as an NMR marker [47]. In addition, the CF_3_ fragment itself has additional advantages, such as increasing the lipophilicity of the compounds. The introduction of a pharmacologically important active pyrazole scaffold that possesses almost all types of pharmacological activities, including anticancer [48,49,50,51], to the molecule can provide additional benefits for the fluorophore. Moreover, the target compound, due to a combination of the donor (pyrazole) [52] and acceptor (phenyl) pyrrole units bonded by the trifluoromethynylene spacer, is characterized by an asymmetric structure and, hence, increased polarization and polarizability of molecules.

The approach to BODIPY **1** includes as a key step the assembly of dipyrromethane **2** from two molecular counterparts: 2,2,2-trifluoro-1-(5-phenylpyrrol-2-yl)ethan-1-ol **3** and pyrrolyl-pyrazole **4** as OH and CH components, respectively (Figure 1). 2,2,2-Trifluoro-1-(5-phenylpyrrol-2-yl)ethan-1-ol **3** was obtained according to the procedure in [53]. For the synthesis of pyrrolyl-pyrazole **4**, the cross-coupling of pyrrole with benzoylbromoacetylene in solid Al_2_O_3_ [54,55,56] was followed by the treatment of 2-benzoylethynylpyrrole **5** thus formed with phenylhydrazine (Figure 1). The formation of pyrrolyl-pyrazole was carried out with slight heating up to 40 °C of 2-benzoylethynylpyrrole with 1.5 (Figure 1) of phenylhydrazine in EtOH overnight.

The reaction of trifluoropyrrolylethanol **3** with pyrrolyl-pyrazole **4** was carried out in the presence of an equimolar quantity of P_2_O_5_ to give dipyrromethane **2** in an 80% yield (Figure 2). The final stage, oxidation of dipyrromethane with DDQ followed by complexation of the formed dipyrromethene with BF_3_ etherate, was realized as a one-pot procedure, to afford the target BODIPY dye **1** in a 39% yield (Figure 2).

### 2.2. Spectroscopic and Photophysical Properties of BODIPY Dye

The electronic absorption and fluorescence spectra of BODIPY **1** in three common organic solvents of different polarities and polarizabilities are shown in Figure 1.

The spectroscopic and photophysical characteristics of dye **1**, including the positions of the maxima of the absorption (λ_abs_) and emission (λ_em_) bands, Stokes shifts (Δν_St_), and fluorescence quantum yields (Φ_F_), are presented in detail in Table 1. The synthesized dye has an intense and narrow long wavelength absorption band situated at λ_abs_ (max) ≥ 610 nm, which is typical for BODIPY derivatives. These bands undergo a red shift by approximately 10 nm as the solvent polarizability increases, on going from MeCN to toluene and DMSO. The fluorescence quantum yields of dye **1** are high (up to 0.9).

To evaluate the photostability of the synthesized dye **1**, long-term irradiation of BODIPY **1** in MeCN and toluene was performed. The experiments indicated the loss of less than 3% of the fluorescence intensity of the sample, implying their photostability (Figure 2).

The same experiments, carried out in pure DMSO, showed slow bleaching of BODIPY **1**, resulting in a visible color change from blue to virtually colorless and the disappearance of the red fluorescence. Probably, this happened due to the specific solute–solvent interaction between fluorophore and DMSO [58]. The exact mode of action of DMSO (and/or H_2_O) and an acid was not studied in this work.

Many applications in medicine and biology, such as fluorescence labeling or sensing, require operation in an aqueous solution. Therefore, studies of the optical properties of BODIPY **1** in H_2_O were performed. Due to the hydrophobic nature of dye **1**, DMSO was used as a co-solvent. The absorption and fluorescence spectra of BODIPY **1** in the DMSO–H_2_O mixture with an incremental amount of water from 0 to 95 vv% are presented at Figure 3a. It is clearly seen that in a mixture of DMSO–H_2_O, a typical aggregation effect [59,60] took place: absorption spectra and fluorescence intensity first remained almost unchanged with increasing water content, and then changed dramatically when a certain content of water was reached. The threshold reflects the starting point of the aggregation of hydrophobic dye. After the aggregation begins, a small increase in water content caused significant fluorescence quenching (Figure 3b). In our case, a significant change in the shape of the absorption spectrum and a considerable decrease in the fluorescence intensity were observed at a water content >50%. At >95% water content, BODIPY **1** practically did not fluoresce.

As in other *meso*-CF_3_-BODIPY dyes with aryl substituents at the positions 3 and 5 [46,47,53,57,61,62,63,64,65], most long-wave S_0_ → S_1_ absorption bands and slightly Stokes-shifted S_1_ → S_0_ respectively to the fluorescence bands, are due to the π-π* one electron excitation from the HOMO to LUMO (Table 2). These band shoulders (Figure 1) could be assigned to vibrational bands of the same transitions [66]. The significantly weaker absorption bands located in the experimental spectra above 450 nm could be assigned to the envelopes of S_0_ → S_2_ and S_0_ → S_3_ transitions of the charge-transfer character (Figure 4). It is of note that there is a large energy gap between the strong main S_0_ → S_1_ transition and the next very weak S_0_ → S_2_ and S_0_ → S_3_ transitions (Table 2). This is consistent with a high experimental Φ_F_ in organic solvents (Table 1) [67].

### 2.3. Evaluation of Lysosome-Specific Sensing

The study of BODIPY **1** as a lysosome tracker was conducted on an Ehrlich ascites carcinoma cell culture. An aqueous BODIPY solution was added to the culture medium, to final concentrations of the substance in samples of 1 μM, 5 μM, 25 μM, and 100 μM.

As a result of the study, it was found that 1.5 h after adding BODIPY **1** solution with a concentration of 1 μM to the cell culture medium, non-localized fluorescence was observed in the whole volume of Ehrlich ascitic carcinoma cells in the red region of the spectrum (extinction filter 550–650 nm) (Figure 5A,B).

After 12 h of incubation, intracellular fluorescence was predominantly observed in rounded vacuolar-type structures, uniformly distributed in the cytoplasmic space, or localized at one of the poles of the tumor cells, which was accompanied by a fluorescence intensity increase. Over the next 12 h, the fluorescence intensity continued to rise, the localization of the dye inside the cells was observed, mainly in the membrane structures, the number of which visualized increased (Figure 5C,D). Upon incubation of carcinoma cells with higher concentrations of the dye (5, 25, and 100 μM) after 1.5 h, along with non-localized fluorescence, colored rounded structures were visualized in the cells (Figure 6, Figure 7 and Figure 8A,B), the largest number of which was observed during incubations with BODIPY at a concentration of 100 μM (Figure 8A,B).

At the same time, all the samples studied completely lacked dye fluorescence in the culture medium. The data obtained indicate that the cell absorption rate of the studied BODIPY in the first hours after its addition, as well as the rate of its transport to intracellular structures, depended on the concentration of the dye in the culture medium. Over the next 12 and 24 h, the number of stained rounded elements in the cells increased, and the fluorescence intensity of intracellular structures also grew in accordance with the concentration of BODIPY introduced (Figure 5, Figure 6, Figure 7 and Figure 8C,D).

The most intense fluorescence was observed in ascites carcinoma cells, when treated with 100 μM BODIPY for 24 h. In this case, the visualized cells’ fluorescence spread to most parts of the cytoplasmic volume (Figure 8C,D), which indicates that the number of intracellular elements stained with the dyes had significantly increased; in addition, the observed fluorescence may have been the result of the fusion of intracellular vacuolar structures, containing BODIPY.

To confirm that the BODIPY-positive structures observed in carcinoma cells were lysosomes, we used immunocytochemical staining of samples, based on the use of antibodies specific for LAMP1 (lysosome-associated membrane protein 1). These proteins, along with LAMP2, are the more abundant components of the glycoconjugate coating on the inner side of the lysosomal membrane, and are involved in such cellular processes as phagocytosis, autophagy, lipid transport, and aging [68,69]. In this regard, these proteins are widely used as lysosome markers in the study of various cell types [70,71,72]. The results of fluorescent staining in our investigation showed that the studied BODIPY **1** dye in the cells of ascitic Ehrlich carcinoma was localized in the same compartments as LAMP1, arranged mainly around the cell nuclei, which was confirmed by the overlapping fluorescent stain areas (Figure 9).

Statistical analysis of the data showed that the differences between the fluorescence intensity in the first 1.5 h of incubation with BODIPY at a concentration of 1 μM did not significantly differ from the control (Figure 10). The control was taken as a weak auto fluorescence of carcinoma cells, visualized with the same filter (550–650 nm) as BODIPY **1**. There were also no significant differences in the fluorescence intensities when comparing the 1.5-h exposure of the 1 μM and 5 μM substance; however, compared with the control, the fluorescence intensity during incubation with BODIPY **1** at a concentration of 5 μM significantly increased. The fluorescence after exposure of 25 and 100 μM for 1.5 h was significantly higher, both in comparison with the control, and when compared with lower concentrations of the dye. At the same time, after 12 and 24 h of incubation, the fluorescence in all samples was significantly more intensive in comparison with the control, thus the increasing of the intensity linearly depended on the concentration of the substance in the medium. However, no significant differences in the fluorescence intensity of samples incubated with 25 μM and 100 μM of BODIPY **1** were observed at later periods. A probable explanation for this fact is that the absorption capacity of a dye by a living cell is limited, and with an increase in its concentration in the medium by four times, there was no significant growth in the fluorescence intensity in the cell, as was observed when exposed to lower concentrations of BODIPY **1**.

A comparison of the data on the changes in fluorescence intensity over time (Figure 10) showed that during incubation with low concentrations (1 and 5 μM), the intensity gradually increased during the day, while application of 25 μM BODIPY **1** led to an abrupt rise in fluorescence intensity in the first 12 h, with a smaller increase in the next 12 h of incubation.

Similar differences in the rate of absorption of the dye by cells during experiment were partially observed when exposed to a 100 μM concentration (Figure 11). The fluorescence intensity during incubation with BODIPY **1** at this concentration in the first hours was already quite high compared to the samples exposed to a lower concentration; however, it increased only in the first 12 h of incubation, whereas no significant growth in intensity was observed in the next 12 h. This may indicate that at higher concentrations, not only the absorption rate of the BODIPY **1** increased, but also the rate of maximum saturation. The inability of the cells to further absorb BODIPY **1** at the 100 μM concentration may have been associated with a transport function interruption, which in turn may have been the result of the progress of the autophagy process in the cells induced by the stressful effect of BODIPY **1** in high concentrations and accompanied by the fusion of lysosomes with autophagosomes and partial degradation of cell structures. Such a scenario also explains the appearance in some cells of larger fluorescently stained structures occupying most of the cytoplasm, the occurrence of which may have been associated with the fusion of lysosomes with each other or with autophagosomes.

## 3. Materials and Methods

### 3.1. General Information

IR spectra were obtained on a “Bruker IFS-25” spectrometer (400–4000 cm^−1^, KBr pellets or thin films on KBr plates). ^1^H (400.1 MHz), ^13^C (100.6 MHz), and ^19^F (376.5 MHz) NMR spectra were recorded on a “Bruker Avance 400” instrument in CDCl_3_. The assignment of signals in the ^1^H NMR spectra was made using COSY and NOESY experiments. Resonance signals of carbon atoms were assigned based on ^1^H-^13^C HSQC and ^1^H-^13^C HMBC experiments. The ^1^H and ^13^C chemical shifts were referenced to CDCl_3_ and ^19^F-CFCl_3_, respectively. The chemical shifts were recorded in ppm. Elemental analyses (C, H, N) were performed on an EA FLASH 1112 Series (CHN Analyzer) instrument. Fluorine content was determined on a SPECOL 11 (Carl Zeiss Jena, Germany) spectrophotometer. Melting point (uncorrected) was determined on a Stuart SMP50 apparatus. High-resolution mass spectra were recorded from acetonitrile solution with 0.1% HFBA on HPLC Agilent 1200/Agilent 6210 TOF instrument equipped with an electrospray ionization (ESI) source.

All solvents (Merck, Darmstadt, Germany, spectroscopic grade) were used without further purification. Photophysical properties were recorded immediately after sample preparation. UV/Vis absorption spectra were measured on a Lambda-35 (Perkin-Elmer, Waltham, MA, USA) spectrophotometer. Excitation and fluorescence spectra were measured on a FLSP-920 combined steady state and time resolved fluorescence spectrometers (Edinburgh Instrument, Levingston, UK). Fluorescence spectra were obtained with a Xenon lamp and 1.0 cm quartz cuvette. All fluorescence and excitation spectra were corrected for the wavelength dependence of the monochromator and the photomultiplier sensitivity. For samples, a right-angle configuration was used, and to avoid re-absorption, the maximum absorbance was kept below 0.1. The temperature of the samples was controlled by an external flow of thermostated water. The fluorescence quantum yields (Φ_F_) of the BODIPY systems were obtained by comparing the areas under the corrected emission spectrum. The Equation (1) was used to calculate quantum yield:Φ_F_ = Φ_ref_ F_sampl_ A_ref_ n^2^ _sampl_/F_ref_ A_sampl_ n^2^ _ref_(1)

The photostability of BODIPY **1** in toluene and MeCN was investigated using a collimated light source from a 150 W Xe lamp in air. The steady-state fluorescence spectroscopy of the samples was monitored for 180 min.

The study of BODIPY **1** as a lysosome sensor was conducted on an Ehrlich ascites carcinoma cell culture, which was obtained from a transplantable culture maintained in white laboratory mice. This culture was acquired from the nursery of Science State Scientific Center for Virology and Biotechnology Vector of the Federal State Institution of the (Russia, Novosibirsk Region, Koltsovo Village), veterinary certificate 254 No. 0336050 of 28 July 2010. Ascitic fluid was taken 9 days after transplantation, since this period is characterized by the logarithmic growth of the Ehrlich ascites carcinoma strain [73]. The resulting cell suspension was diluted 500 times with DMEM culture medium (Dulbecco’s Modified Eagle’s Medium, Sigma-Aldrich Co, St. Louis, MO, USA) supplemented with 10% FBS (Fetal Bovine Serum, Sigma-Aldrich, Co, St. Louis, MO, USA) and 1% antibiotic/antimycotic (10.000 units/mL penicillin, 10,000 μg/mL streptomycin and 25 μg/mL amphotericin B, Gibco (Thermo Fisher Scientific, Waltham, MA, USA). Cell culture was cultured for 1 day at a temperature of 37 °C, a humidity of 80%, and a CO_2_ content of 5% under the conditions of a CO_2_ incubator in BioStation CT (Nikon, Nikon Corporation, Tokyo, Japan).

The BODIPY **1** was dissolved in DMSO, followed by adjustment to a final concentration with bi-distilled water (the total dilution of DMSO was about 20 times). An aqueous BODIPY solution was added to the culture medium, to final concentrations of the substance in samples of 1 μM, 5 μM, 25 μM, and 100 μM. From the moment the BODIPY solution was added, the cell culture was incubated for 24 h.

Cells for immunofluorescence studies were fixed with 70% ethanol and, after permeabilization, with 1% Triton X-100, and stained with antibodies to LAMP1 (Cat. MA5-29385, Invitrogen, Carlsbad, CA, USA) in a dilution of 1:200. Alexa fluor 488 goat anti-rabbit IgG (H + L) (Cat. NA-11034, Invitrogen, Carlsbad, CA, USA) in a dilution of 1:300 was used as secondary antibody. Nuclei were stained with Hoechst (Cat. NH-3570, Invitrogen, Carlsbad, CA, USA), 1:300.

Visualization of the specific staining and photofixation of samples 1.5 h, 12 h, and 24 h after adding the substance to the medium was carried out using BioStation CT 4.1, Nikon, Nikon Corporation, Tokyo, Japan). Fluorescence intensity analysis was performed applying the NIS-Elements AR, software product v. 5.00. The study was carried out using equipment from the centers of collective use “Diagnostic images in surgery”. Statistical analysis was performed in the R programming environment. The Kruskal–Wallis criterion was used for the variance analysis, and a posteriori analysis was performed using the Tukey criterion.

### 3.2. Synthesis of 1,5-Diphenyl-3-(1H-pyrrol-2-yl)-1H-pyrazole (***4***)

An excess of phenylhydrazine (0.162 g, 1.5 mmol) was added to the solution of 2-benzoylethynylpyrrole (0.195 g, 1.0 mmol) in EtOH (10 mL). The reaction mixture was stirred at 40 °C overnight, diluted with 30 mL of H_2_O, and extracted by diethyl ether. The residue after removing the solvent was recrystallized from *n*-hexane, to afford 0.317 g (73%) of pyrrolyl-pyrazole as a yellow viscous oil.

^1^H NMR (400.13 MHz, CDCl_3_) δ 9.08 (br s, 1H), 7.35–7.30 (m, 7H), 7.27–7.25 (m, 3H), 6.84–6.82 (m, 1H), 6.65 (s, 1H), 6.56–6.54 (m, 1H), 6.29–6.27 (m, 1H).

^13^C NMR (100.6 MHz, CDCl_3_) δ 146.2, 144.3, 140.1, 130.5, 129.0 (2C), 128.8 (2C), 128.5 (2C), 128.4, 127.4, 125.6, 125.3 (2C), 118.5, 109.3, 106.8, 104.3.

IR (KBr, cm^−1^): 3439, 3257, 3061, 1599, 1541, 1497, 1451, 1409, 1356, 1179, 1108, 1071, 1028, 979, 908, 799, 763, 729, 697.

Anal. Calcd for C_19_H_15_N_3_: C, 79.98; H, 5.30; N, 14.73; Found: C, 80.21; H, 5.48; N, 14.54%.

#### 3.2.1. Synthesis of 1,5-Diphenyl-3-(5-(2,2,2-trifluoro-1-(5-phenyl-1*H*-pyrrol-2-yl)ethyl)-1H-pyrrol-2-yl)-1*H*-pyrazole (**2**)

P_2_O_5_ (0.170 g, 1.2 mmol) was added to the stirred mixture of pyrrolyl-pyrazole **3** (0.285 g, 1.0 mmol) and trifluoro(pyrrolyl)ethanol **2** (0.241 g, 1.0 mmol) in dried dichloromethane (6 mL), and the reaction mixture was stirred at room temperature for 3 h. Then NaHCO_3_ (0.202 g, 2.4 mmol) was added and the mixture was stirred at room temperature for 10 min. The formed precipitate was filtered off, and the residue after removing the solvent was purified with column chromatography on SiO_2_ (*n*-hexane/Et_2_O 3:1) to afford 0.406 g (80%) of dipyrromethane as a white solid. Mp 78–80 °C.

^1^H NMR (400.1 MHz, CDCl_3_) δ 9.04 (br s, 1H), 8.36 (br s, 1H), 7.43–7.42 (m, 2H), 7.36- 7.29 (m, 10H), 7.25–7.19 (m, 3H), 6.66 (s, 1H), 6.54–6.53 (m, 1H), 6.47–6.46 (m, 1H), 6.35–6.31 (m, 2H), 4.91–4.85 (q, *J* = 9.0 Hz, 1H).

^13^C NMR (100.6 MHz, CDCl_3_) δ 145.6, 144.7, 134.0, 133.0, 132.5, 130.4, 129.1 (2C), 128.9 (2C), 128.8 (2C), 128.6 (2C), 128.5, 127.7, 126.7, 126.5, 125.4 (2C), 125.2 (q, *J* = 280.0 Hz), 124.1 (2C), 124.0, 123.8, 110.9, 110.3, 107.4, 106.7, 104.4, 43.6 (q, *J* = 30.2 Hz).

IR (KBr, cm^−1^): 3346, 3037, 1601, 1546, 1503, 1451, 1406, 1351, 1255, 1162, 1107, 1044, 909, 763, 733, 698, 523.

Anal. Calcd for C_31_H_23_F_3_N_4_: C, 73.22; H, 4.56; F, 11.21; N, 11.02; Found: C, 73.54; H, 4.71; F, 11.43; N, 10.85%.

#### 3.2.2. Synthesis of 4,4-Difluoro-5-phenyl-3-(1,5-diphenyl-3-1H-pyrazolyl)-8-trifluoromethyl-4-bora-3a,4a-diaza-s-indacene (**1**)

The mixture of dipyrromethane (0.508 g, 1.0 mmol) and DDQ (0.227 g, 1.0 mmol) in dried dichloromethane (10 mL) was stirred at room temperature for 1 h. The (*i*-Pr)_2_NEt (1.29 g, 10.0 mmol) was added, the solution was stirred for 10 min and cooled to 0 °C, then boron trifluoride etherate (1.562 g, 11.0 mmol) was added dropwise. The reaction mixture was stirred at 0 °C for 2 h, then the ~2/3 of solvent was removed under vacuum, and the obtained residue was purified by column chromatography (SiO_2_, hexane/dichloromethane, 2:1) to yield 0.216 g (39%) of the BODIPY dye as green crystals, mp 175–177 °C.

^1^H NMR (400.1 MHz, CDCl_3_) δ 7.94 (m, 2H), 7.58 (s, 1H), 7.47 (m, 5H), 7.40 (m, 1H), 7.33 (m, 8H), 7.26 (m, 1H), 7.25 (m, 1H), 6.72 (m, 1H).

^13^C NMR (100.6 MHz, CDCl_3_) δ 159.9, 155.1, 145.0, 143.5, 139.7, 134.4, 132.9, 132.4, 130.4, 130.0, 129.9, 129.7 (t, *J* = 4.0 Hz), 129.2 (2C), 129.1 (2C), 128.9, 128.7, 128.6 (2C), 128.5 (2C), 128.4, 128.3, 125.4 (2C), 125.3 (q, *J* = 32.8 Hz), 123.3, 122.8 (q, *J* = 275.9 Hz), 122.1, 111.3 (t, *J* = 4.0 Hz).

^19^F NMR (376.5 MHz, CDCl_3_) δ −54.9 (CF_3_), −136.0 (BF_2_).

IR (KBr, cm^−1^): 1574, 1521, 1472, 1397, 1351, 1279, 1224, 1137, 1086, 1020, 919, 797, 760, 692.

Anal. Calcd for C_31_H_20_BF_5_N_4_: C, 67.17; H, 3.64; B, 1.95; F, 17.14; N, 10.11; Found: C, 67.26; H, 3.90; F, 17.32; N, 9.89%.

HRMS (ESI) *m*/*z*: [M + H] + Calcd for C_31_H_20_BF_5_N_4_ 555.1794; Found 555.1783.

## 4. Conclusions

A new unsymmetrical *meso*-CF_3_-BODIPY fluorescent dye, with phenyl and pyrazolyl substituents emitted at λ_em_ ≈ 640–650 nm with high quantum yield (0.7–0.9), was proposed for fluorescence visualization of lysosomes. This BODIPY **1** is characterized by the ability to accumulate in cell lysosomes, with the rate of its absorption by the cell and intracellular transport, as well as the rate of saturation of the cell with the substance, depending on its concentration in the culture medium. The observed absorption limit of the substance by cells at a concentration of BODIPY **1** in the culture medium may be associated with transformations induced in cells by the stressful effect of a high BODIPY concentration. As a result of the analysis of the obtained data, the concentration of 5 μM was chosen as the most convenient concentration of the tested substance for fluorescent staining of lysosomes, since during incubation with such a concentration intense fluorescent staining was observed, gradually increasing during the day, and at the same time there was no toxic effect on cells, as when using high concentrations of BODIPY **1**. Thus, can be concluded that the synthesized BODIPY is easy to use, stable in the form of an aqueous solution, does not require special storage conditions, and is suitable for staining intracellular structures.

## Data Availability

Not applicable.

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
