# Peer review of "Application of *meso*-CF_3_-Fluorophore BODIPY with Phenyl and Pyrazolyl Substituents for Lifetime Visualization of Lysosomes"

_molecules, 2022, doi:10.3390/molecules27155018_

Round 1

Reviewer 1 Report

In this manuscript, Trukhan et al. synthesized a meso-CF3-fluorophore BODIPY with phenyl and pyrazolyl substituents and used it in the visualization of lysosomes. However, there are many major flaws in the manuscript and the conclusions are overstated. Moreover, the novelty and significance of the contents are not clear. I don’t think this manuscript is ready for publication.

1.     The rationale for the design of the BODIPY has not been mentioned at all. Why introduce a trifluoromethyl group at the meso-position? Why introduce a pyrazolyl group? It seems to be a random study of BODIPY modification. Many related meso-CF3-BODIPY should be commented in the manuscript.

2.     There are many errors in the synthetic conditions, including many strange upper-case letters.

3.     The lysosome-specific sensing study using the BODIPY is not accurate. First, a confocal microscope should be used in the study. Figure 7 is not clear, which showed that LAMP1 has better staining than BODIPY. Second, the quantitative study should be the average fluorescence intensity per cell. Third, a cell viability study should be carried out before discussing the toxicity of the BODIPY.

4.     In lines 119-128, the logic is confusing and the author writes that it is of note that there is a large energy gap between the strong main S0→S1 transition…This indicates that dye 1 should have high Φf. I don't understand how to conclude that dye 1 has a higher fluorescence quantum yield. As far as I know, a smaller energy level difference from HOMO to LUMO means easier excitation, which has no relation to fluorescence emission quantum yield.

5.     There is an error in line 113, where the polarity of MeCN is greater than that of toluene?

6.     The authors state that the fluorescence intensity is proportional to the concentration, however in Figures 4, 5, and 6, the fluorescence intensity is stronger for 1.5 h incubation at 5 µM than for 1.5 h incubation at 25 µM and 100 µM. Also, the number of cells in Figure 5 is significantly less than in Figures 4 and 6, which I don't think can be compared to the other concentrations. It should be the average fluorescence intensity per cell.

Author Response

Reviewer 1 wrote:

  1. The rationale for the design of the BODIPY has not been mentioned at all. Why introduce a trifluoromethyl group at the meso-position? Why introduce a pyrazolyl group? It seems to be a random study of BODIPY modification. Many related meso-CF3-BODIPY should be commented in the manuscript.

Answer: Following the advice of the respected reviewer, we have added the following text to manuscript:

The choice of this compound is due to the fact that BODIPY fluorophores containing aromatic or heterocyclic substituents at the 3- and (or) 5-positions [38, 39] or the strongly electron-withdrawing substituents, such as CF3, at the meso-position of BODIPY core cause a deep bathochromic shift of absorption and emission maxima and therefore can be applied in biochemical studies. For example, meso-CF3-BODIPY dyes are employed as fluorescent sensors for in vivo imaging systems and in photodynamic therapy [40-46]. The BODIPY dyes with meso-CF3-group also have at least two benefits as a biochemical probe. First, the probe molecule is small. Second, the CF3-group is known to be useful as an NMR marker [47]. In addition, CF3 fragment itself has additional advantages, such as increasing the lipophilicity of the compounds. The introduction of a pharmacologically important active pyrazole scaffold that possesses almost all types of pharmacological activities, including anticancer [48-51] to the molecule can provide additional benefits for fluorophore. Moreover, the target compound due to a combination of the donor (pyrazole) [52] and acceptor (phenyl) pyrrole units bonded by trifluoromethynylene spacer is characterized by an asymmetric structure and, hence, increased polarization and polarizability of molecules.

  1. There are many errors in the synthetic conditions, including many strange upper-case letters.

Answer: Schemes 1 and 2 are corrected.

  1. The lysosome-specific sensing study using the BODIPY is not accurate. First, a confocal microscope should be used in the study. Figure 7 is not clear, which showed that LAMP1 has better staining than BODIPY. Second, the quantitative study should be the average fluorescence intensity per cell. Third, a cell viability study should be carried out before discussing the toxicity of the BODIPY.

Answer:

Since, unfortunately, we do not have the opportunity to use a confocal microscope, the study was carried out on the available equipment, that is, at a BioStation that allows obtaining images directly on the incubator condition after live preparation fluorescent staining. This approach made it possible to analyze the test substance accumulation for a given period of time not only in the same preparations, but also in the same cells, as can be seen from images 3-6.

In Figure 7, indeed, BODIPY came to look less bright after the procedures of fixation and multi-stage antibody stain, which includes several sample washing steps. However, in this case, it can be observed that even after these procedures, the dye is not only still registered in cells, but also allows one to estimate its localization, including in comparison with LAMP1. Such stability also emphasizes the uniqueness of the resulting substance.

The average intensity data presented in the study was obtained for each fluorescent spot, and depending on the staining specificity, the program analyzes the fluorescence intensity of the whole cell or individual structures included in the cell. At least 10 images were analyzed for each concentration and time interval, which allows us to obtain good statistical results.

Since this study was not originally focused on the toxicity of this substance to cells, but on the possibility of using BODIPY for intravital staining of lysosomes, including analysis of its uptake by cells and its distribution within cells, a toxicity study was not carried out. However, such a study is planned in the future as part of further study of this BODIPY’s properties. The presented preliminary conclusions about the toxicity of high concentrations, although they are in the nature of an assumption, in this case, according to the authors, are concordant with the behavior of the substance in carcinoma cells and the observed effects.

  1.   In lines 119-128, the logic is confusing and the author writes that it is of note that there is a large energy gap between the strong main S0→S1 transition…This indicates that dye 1 should have high Φf. I don't understand how to conclude that dye 1 has a higher fluorescence quantum yield. As far as I know, a smaller energy level difference from HOMO to LUMO means easier excitation, which has no relation to fluorescence emission quantum yield.

Answer:

 Probably, the reviewer's remark concerns the S0-S1 gap and its influence on the value of the fluorescence quantum yield. However, in this paper, we are talking about the energy gap between the first allowed state S1 (oscillator strength f, f = 0.79) and the second, forbidden excited state with charge transfer S2 (f = 0.06), i.e., the gap S1-S2. Note, that if this gap is small (< 0.2 eV), then it is very probable that the fluorescent state S1 will be “quenched” (due to the electronic-vibrational interaction with S2), and in such systems one should expect a relatively small ΦF.Our calculations show that there is no such quenching channel for BODIPY 1 [there is a large energy gap S1-S2 (1.3 eV)]. This fact is in agreement with the high experimental values ​​of ΦF for BODIPY 1 and is an argument in favor of the correctness of the calculations performed.

 The phrase “This is an indicator that dye 1 should have high Ff in line with experimental date in Table 1 in organic solvents.” was changed on phrase “This is consistent with high experimental FF in organic solvents (Table 1).”

  1. There is an error in line 113, where the polarity of MeCN is greater than that of toluene?

Answer: In line 113 it talks about polarizability (n2), not polarity of solvents.

  1. The authors state that the fluorescence intensity is proportional to the concentration, however in Figures 4, 5, and 6, the fluorescence intensity is stronger for 1.5 h incubation at 5 µM than for 1.5 h incubation at 25 µM and 100 µM. Also, the number of cells in Figure 5 is significantly less than in Figures 4 and 6, which I don't think can be compared to the other concentrations. It should be the average fluorescence intensity per cell.

Answer:

No matter how the images look when examined with the unaided eye, the staining intensity analysis performed by the program indicates that already in the first 1.5 hours the fluorescence intensity is significantly higher when cells are incubated with higher concentrations of BODIPY (25 µM and 100 µM), which is shown in Figure 8.

Reviewer 2 Report

The synthesis and application of a bright far-red emitting unsymmetrical meso-CF3-BODIPY fluorescent dye with phenyl and pyrazolyl substituents were reported by Irina S. Trukhan et al. The result is very interesting for the researcher working on the field of BODIPY. It is thus well suited for publication in this journal. However, minor changes are needed. Before publication, the following points should be addressed:

1. Based on the authors' previous work, there is little difference in the properties and synthesis difficulty of the symmetric and asymmetric BODIPY derivatives of meso-position CF3. It is recommended to further elaborate on the reasons for the design synthesis and further application of asymmetric BODIPY 1 in this paper.

1. In the introduction part, the following recent references about the advantage of BODIPY and their application might be included in the revised manuscript: 10.1016/j.saa.2020.119199,10.1021/acsaem.2c01320

2. Some comments about the molecular design this work should be added in the introduction.

3. There are many errors in the Scheme 1 which need be revised. For example, 0.5 h should be instead of 0,5 h.

4.  The molar extinction coefficients in different solvents are all an integer, and they are strange.

5.  High resolution mass spectrometry, and photostability tests on BODIPY 1 should be performed.

6. Φf should be ΦF.

7. Some awkward sentences in this work need to be corrected and rephrased.

Author Response

Reviewer 2 wrote:

  1. Based on the authors' previous work, there is little difference in the properties and synthesis difficulty of the symmetric and asymmetric BODIPY derivatives of meso-position CF3. It is recommended to further elaborate on the reasons for the design synthesis and further application of asymmetric BODIPY in this paper.

Answer:

Indeed, based on the technology we developed, we were able to synthesize both symmetrical and asymmetrical BODIPY fluorophores. But, since asymmetric fluorophores containing both donor and acceptor substituents should have increased polarization and polarizability of molecules, BODIPY 1 was chosen by us as object of our study. Phrase “Moreover, the target compound due to a combination of the donor (pyrazole) and acceptor (phenyl) pyrrole units bonded by trifluoromethynylene spacer is characterized by an asymmetric structure and, hence, increased polarization and polarizability of molecule.” was added to the section ”Results and discussion” of the manuscript.

  1. In the introduction part, the following recent references about the advantage of BODIPY and their application might be included in the revised manuscript: 10.1016/j.saa.2020.119199,10.1021/acsaem.2c01320

Answer:

Papers proposed by reviewer are included in the introduction as references 3 and 12.

  1. Some comments about the molecular design this work should be added in the introduction.

Answer:

The comments about the molecular design were added in section “Results and discussion”

  1. There are many errors in the Scheme 1 which need be revised. For example, 0.5 h should be instead of 0,5 h.

Answer:

Schemes 1 and 2 were corrected.

  1. The molar extinction coefficients in different solvents are all an integer, and they are strange.

Answer:

The molar extinction coefficients were determined with accuracy +/- 200 M-1cm-1. This phrase was added as notes to Table 1.

  1. High resolution mass spectrometry of BODIPY should be performed.

Answer:

Data of high resolution mass spectrometry of BODIPY were added ton the manuscript.

  1. Photostability tests on BODIPY should be performed.

Answer:

 Photostability tests on BODIPY in toluene and MeCN were performed and the results were added to the paper. The solution of BODIPY 1 in DMSO is chemically unstable, so photostability was not tested in this solvent.

The text “The photostability of BODIPY 1 in toluene and MeCN was investigated using a collimated light source from a 150 W Xe lamp in air. The steady-state fluorescence spectroscopy of the samples was monitored for 180 min” was added in section ‘Materials and method”.

The text: “To evaluate photostability of synthesized dye 1, long-term irradiation of BODIPY 1 in MeCN and toluene has been performed. The experiments indicate the loss of less than 3% of fluorescence intensities of the samples, implying their photostability (Figure 2).  The same experiments, carried out in pure DMSO, show slow bleaching of BODIPY 1, resulting in a visible color change from blue to virtually colorless and a disappearance of the red fluorescence. Probably it happens due to specific solute–solvent interaction between BODIPY 1 and DMSO [58]. The exact mode of action of DMSO (and/or H2O) and an acid has not been studied in this work” and Figure 2 were added in section “Results and discussion”.

  1. Φfshould be ΦF.

Answer:

Corrected as recommended.

  1. Some awkward sentences in this work need to be corrected and rephrased.

Answer:

The text has been carefully revised again and changed in some places.

Author Response

Reviewer 3 wrote:

  1. Lack of explanation in probe design.

The point of the use of meso-CF3-BODIPY fluorescent dye 1 with phenyl and pyrazolyl substituents can be specifically strained for live cell lysosomes, but the point was not fully discussed in the present paper. First, the results are not shown in comparison with non- or different substituent meso-CF3-BODIPYs, so the merit of the phenyl and pyrazolyl substituent groups is not clear.

Answer: Following the advice of the respected reviewer, we have added the following text to manuscript:

The choice of this compound is due to the fact that BODIPY fluorophores containing aromatic or heterocyclic substituents at the 3- and (or) 5-positions [38, 39] or the strongly electron-withdrawing substituents, such as CF3, at the meso-position of BODIPY core cause a deep bathochromic shift of absorption and emission maxima and therefore can be applied in biochemical studies. For example, meso-CF3-BODIPY dyes are employed as fluorescent sensors for in vivo imaging systems and in photodynamic therapy [40-46]. The BODIPY dyes with meso-CF3-group also have at least two benefits as a biochemical probe. First, the probe molecule is small. Second, the CF3-group is known to be useful as an NMR marker [47]. In addition, CF3 fragment itself has additional advantages, such as increasing the lipophilicity of the compounds. The introduction of a pharmacologically important active pyrazole scaffold that possesses almost all types of pharmacological activities, including anticancer [48-51] to the molecule can provide additional benefits for fluorophore. Moreover, the target compound due to a combination of the donor (pyrazole) [52] and acceptor (phenyl) pyrrole units bonded by trifluoromethynylene spacer is characterized by an asymmetric structure and, hence, increased polarization and polarizability of molecules.

  1. Second, the spectroscopic and photophysical characteristics of BODIPY 1 in aqueous solution, which is one of the important factors in the practical use of the dye, is not described. Although dye 1 has a high quantum yield (ca. 0.7–0.9) in organic solvent.

Answer:

This text and corresponding Figure were added in manuscript:

Many applications in medicine and biology, such as fluorescence labeling or sensing require operations in an aqueous solution. Therefore, the studies of the optical properties of BODIPY 1 in H2O have been performed. Due to the hydrophobic nature of dye 1, DMSO has been used as a co-solvent. Absorption and fluorescence spectra of BODIPY 1 in DMSO - H2O mixture with an incremental amount of water from 0 to 95 vv% are presented at Fig. 3a. It is clearly seen that in a mixture of DMSO - H2O, a typical aggregation effect [59, 60] takes place: absorption spectra and fluorescence intensity first remain almost unchanged with increasing water content, and then change dramatically when certain content of water is reached. Threshold reflects the starting point of an aggregation of hydrophobic dye. After the aggregation begins, a small increase in water content causes significant fluorescence quenching (Fig. 3b). In our case, a significant change in the shape of the absorption spectrum and a considerable decrease in the fluorescence intensity are observed at a water content > 50%. At >95% water content, BODIPY 1 practically does not fluoresce.

  1. The authors claim that the dye “can be used for the lifetime visualization of lysosomes in live cells”. However, Figures 3–7 are of poor quality, not enough at all to demonstrate the specificity of the probe for live cell imaging. It seems the fluorescence signals in those images are evenly distributed in cytoplasm without focused localization, especially, when dye 1 is used at high concentrations (25 and 100 μM). These results may indicate that the excess probe is too hydrophobic or sticky to remove under live cell settings and non-specifically binds to intracellular proteins, leading to high background and precluding the specific imaging of lysosomes in live cells. To clarify this issue, the authors should investigate the feasibility of using dye 1 as a lysosome marker by performing a time-lapse colocalization study employing the LysoTracker probes as the standard lysosome markers in live cells. The colocalization of dye 1 with LysoTracker probe will be quantified by Pearson’s coefficients.

Answer:

To study the localization of the synthesized BODIPY in carcinoma cells antibodies specific to the lysosome membrane protein were chosen, rather than LysoTracker, in order to avoid competition of substances that may act in a similar manner and probably use the same mechanisms of transport into cell lysosomes and, accordingly, to avoid possible false conclusions that may result from changes in absorption and accumulation in the such competition condition.

The alleged “non-specificity” of this substance is not discussed by the authors, since when used at low concentrations, rounded structures are clearly visible within the cells on the obtained images, which makes it possible to suggest that we cannot talk about simple binding to intracellular proteins. Similar non-localized intracellular staining was observed by the authors in the study of a number of other compounds, also belonging to the BODIPY class at different concentrations (data not presented in this work), and the data obtained are highly different from those for BODIPY 1 described in the article, which indicates the unique properties of this compound.

  1. Addressing dye performance compared to available LysoTracker probes.

Since dye 1 structure does not include a basic amino group, the intriguing question is where dye 1 would be accumulated in cellular lysosomes. The authors should propose a reasonable mechanism how dye 1 localized in to cellular lysosomes. Furthermore, the authors should investigate that whether this probe has the ability to label fixed cells, permeabilized cells, and NH4Cl-treated cells, compare to LysoTracker probes, which does not label efficiently under the same conditions. On the other hand, lysosome tracking probes should possess the ability to stay in lysosomes for a long period of time. Unfortunately, commercially available LysoTracker probes have some limitations: (i) prolonged accumulation of LysoTracker probes in the intracellular environment increases the cellular pH, causing fluorescent dye quenching as well as physiological and 2 morphological changes in the lysosomes due to the “alkalinizing effect” and (ii) the low photostability limits their use for long term tracking of lysosomes to observe dynamic changes in lysosomal morphology in a stipulated time. I recommended the authors to demonstrate that dye 1 can track lysosomes in live cells for at least three days and demonstrate stable imaging of most physiological activities of lysosomes in live cells.

Answer:

The mechanism of transport into lysosomes, as well as other possible properties of this compound are the object of further research. The BODIPY ability to stain fixed samples, including after permeabilization, was tested by the authors, and data are not presented in this study because staining was completely absent in both cases. In this regard, this substance is recommended by the authors exclusively as an intravital dye. The maximum duration of staining is also an object of further research, however, at this stage, the authors can estimate that, firstly, the time period during which this dye is visualized in cells exceeds 7 days, and secondly, the stain is still registered in cells after procedures of sample fixation and a multi-stage antibody staining including several sample washing steps. In this connection, the authors consider this BODIPY to have unique properties, and the fact that this compound can become the object of numerous subsequent studies including those proposed in this review in our opinion can be attributed to its undeniable advantages and indicate that it may be of interest to a wide range of researchers.

  1. Minor point to consider in the following issue:

Please check NMR spectra of the compounds

e.g. compound 1: 1H NMR (400.1 MHz, CDCl3) δ 7.94 (m, 2H), 7.58 (s, 1Н), 7.47 (m, 5Н),

7.40 (m, 1Н), 7.33 (m, 8Н), 7.26 (m, 1Н), 7.25 (m, 1Н), 6.72 (m, 1Н).

Answer:

NMR spectra of the compounds were checked

Round 2

Reviewer 1 Report

The authors have addressed my concerns. It may be accepted in the current form.
